# OncoNLP: Cancer Comprehend Annotation – a pipeline for cancer phenotype and clinical extraction

Thanh Duong
*Department of Computer Science and Engineering*
*University of South Florida*
Tampa, FL
thanh.duong@moffitt.org

Thanh Thieu
*Department of Machine Learning*
*Moffitt Cancer Center*
Tampa, FL
thanh.thieu@moffitt.org

*Abstract*—Information extraction from clinical text is needed to comprehend patient conditions and determine anticancer treatment. Existing NLP systems, such as Amazon Comprehend Medical, Clamp, and DeepPhe, are proprietary and suboptimal due to the shift in the textual distribution and expression of cancer phenotypes. We introduce OncoNLP, a natural language processing toolkit that comprises deep neural network BERT models designed to extract cancer phenotype and related biomedical information. Currently, the toolkit contains two primary components: a Biomedical BERT(BiomedBERT) model to extract general medical information and a Cancer Bert(CaBERT) model to identify the primary tumor site and histology. We evaluate the performance of BiomedBERT on Informatics for Integrating Biology & the Bedside (i2b2) dataset against Amazon Comprehend Medical and Clamp. BiomedBERT outperforms all other methods with an exact matching F1-score of 88.5%. Next, we evaluate CaBERT on a Moffitt dataset with over 2000 clinical notes against DeepPhe, which shows a 50% improvement in both tumor site and histology tasks. Lastly, we introduce a knowledge prompt with OncoNLP and engineer it for large language models. We name it OncoNLP-Assist, a chatbot system powered by OncoNLP and Llama2 that could extract information from electrical health records and interact with physicians.

*Index Terms*—Oncology extraction, pathology report informatics

## I. INTRODUCTION

Pathology reports provide comprehensive documentation of biological samples, which is crucial for advancing cancer research through large-scale data analysis. These reports are vital in clinical trial screening, case identification, prognostication, surveillance, treatment selection, and numerous other applications [1]. For extracting clinical information from pathology reports, Natural language processing (NLP) enables the detection and examination of critical data points, facilitating better understanding and utilization of the information contained within these records. These advancements in NLP have significantly improved the efficiency and accuracy of data extraction from clinical documents, thereby enhancing the overall quality of healthcare delivery and research. However, automated extraction of information from pathology reports can prove challenging because of the wide-ranging diversity in language usage and documentation formats [2]. The challenges include a range of issues, such as the complex organization of pathology ontology, the presence of clinical diagnoses accompanied by detailed explanations, varying terminology used to describe the same cancer, as well as synonymous and unclear terms, and the occurrence of multiple diagnoses within a single report.

Several popular NLP systems have been developed to process pathology text, including text analysis and knowledge extraction systems such as Amazon Comprehend Medical [3], Clamp [4], and DeepPhe [5]. These systems can extract diverse cancer phenotypes and have been successfully applied to numerous information extraction tasks, including identifying tumor characteristics, staging information, and treatment outcomes. By leveraging advanced NLP techniques, these tools facilitate a more efficient and accurate analysis of pathology reports, ultimately enhancing research and clinical decision-making in oncology. Despite their success, these NLP systems each have weaknesses preventing them from effectively and safely applying to cancer treatment. For instance, DeepPhe is a rule-based system that can struggle with the flexibility required for diverse clinical narratives. Next, Clamp utilizes Conditional Random Fields (CRF) [6]and dictionary-based methods, which may lack the adaptability to handle the nuances and complexities of varying clinical terminologies and contexts. Lastly, Amazon Comprehend Medical is limited to online access, raising significant security concerns as it is not designed to handle Protected Health Information (PHI) safely. Therefore, these limitations highlight the necessity for robust and secure NLP solutions designed specifically for oncology data's complex and sensitive nature.

Recently, attention-based Encoder-Decoder architectures in pathology text mining have been proposed to overcome some of the limitations of rule-based techniques [7]. These architectures utilize the latest developments in NLP techniques to enhance the comprehension of contextual relationships, providing a more flexible and accurate approach to analyzing clinical text. For example, the BERT model [8] is a language representation model that provides contextualized word embeddings. It relies on a multi-layer bidirectional encoder, and instead of employing sequential recurrence, it utilizes parallel attention layers in the transformer neural network. Hence, the BERT model can depict words or sequences in a manner that captures contextual details, resulting in distinct representations of the same sequence of words when encountered in diverse contexts.

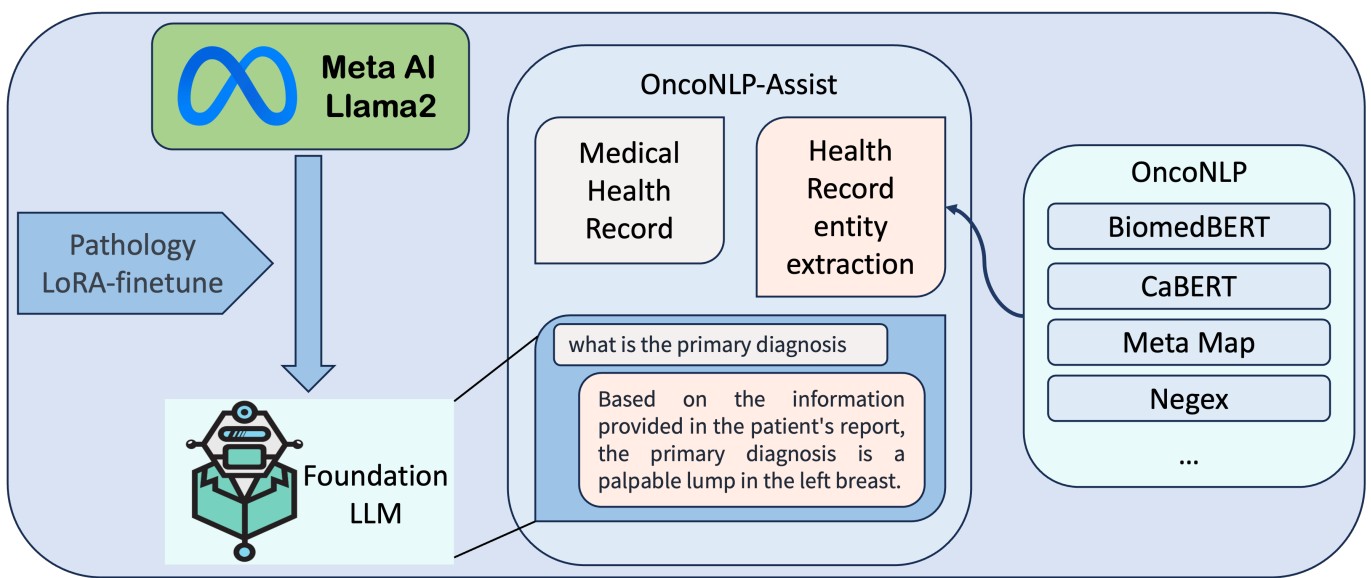

Fig. 1. The architecture of OncoNLP-Assist integrates insights from both the OncoNLP pipeline and the Llama2 model for enhancing its knowledge base.

In this work, we propose OncoNLP and OncoNLP-Assist, a natural language processing toolkit that comprises deep neural network Transformers models designed to extract cancer phenotype and related biomedical information. In the OncoNLP pipeline, we design each component model as a module that could easily be upgraded or substituted with a higher-performing model. In our experiment, we propose BiomedBERT - a pre-trained masked language model trained on the MIMIC-III dataset and 349,544 pathology cancer reports. We train it using the same structure as clinicalBERT [9] and then fine-tune it for downstream tasks with the i2b2 [10] dataset to extract name tag entities, including Problem, Test, and Treatment. OncoNLP also contains several modules such as CaBERT [11] for extracting site and histology code, Metamap [12] for extracting body location, and Negex [13] for extracting negation relation. Next, we introduce OncoNLP-Assist, a pathology assistant that integrates the comprehensive knowledge of OncoNLP with the Llama2 [14] model. This combination enables the extraction of essential information from health records and facilitates interaction with physicians. The Llama2 model is fine-tuned with Low-Rank Adaptation (LoRa) on the pathology cancer reports and serves as a physician's assistant for answering any question according to the health report. The architecture of OncoNLP-Assist is shown in Figure 1.

## II. RELATED WORK

Several notable NLP systems have been developed for processing pathology text, including text analysis and knowledge extraction tools like Amazon Comprehend Medical [3], Clamp [4], and DeepPhe [5]. These systems are proficient in extracting a wide range of cancer phenotypes and have been effectively utilized in various information extraction tasks such as identifying tumor characteristics, staging information, and treatment outcomes. These tools enable more efficient and

accurate analysis of pathology reports by leveraging advanced NLP techniques, thus enhancing research and clinical decision-making in oncology.

However, each of these NLP systems has limitations that restrict their effective and safe application in cancer treatment. For instance, DeepPhe is a rule-based system that lacks the flexibility needed to handle diverse clinical narratives. Rule-based systems like DeepPhe rely on predefined sets of rules and patterns to extract information. These systems can be effective for specific tasks but often fail to generalize to the varied and complex language used in clinical texts. This rigidity means that any deviation from the expected format can lead to missed or incorrect information extraction, limiting the system's overall applicability and accuracy. Similarly, Clamp employs Conditional Random Fields (CRF) and dictionary-based methods to perform information extraction. While CRF models can capture sequential dependencies in text, they often struggle with the intricate and context-dependent nature of clinical terminologies. Dictionary-based methods, although useful for identifying specific terms, cannot adapt to new or evolving medical language and fail to understand the broader context in which terms are used. This limitation can result in inaccurate information extraction, particularly in cases where nuanced understanding is crucial for clinical decision-making. Additionally, Amazon Comprehend Medical's requirement for online access poses significant security concerns. As a cloud-based service, it requires transmitting potentially sensitive health information over the internet. This raises issues related to data privacy and compliance with regulations such as the Health Insurance Portability and Accountability Act (HIPAA). The service is not specifically designed to handle Protected Health Information (PHI) securely, which increases the risk of data breaches and unauthorized access. These security concerns make it unsuitable for applications that demand

stringent data protection measures, particularly in healthcare settings.

Recently, attention-based Encoder-Decoder architectures have been introduced in pathology text mining to overcome some of the shortcomings of rule-based techniques. These architectures leverage the latest advancements in NLP to enhance the understanding of contextual relationships, offering a more flexible and accurate method for analyzing clinical texts. For example, the BERT model generates contextualized word embeddings using a multi-layer bidirectional encoder and parallel attention layers in the transformer neural network, rather than sequential recurrence. This allows BERT to capture contextual nuances, resulting in unique representations of the same sequence of words in different contexts.

In this work, we propose OncoNLP and OncoNLP-Assist, a natural language processing toolkit comprising deep neural network Transformer models designed to extract cancer phenotypes and related biomedical information. OncoNLP-Assist builds upon the flexibility of Transformer models, allowing it to handle diverse clinical text with high accuracy. In our experiments, we utilized BiomedBERT and a chat model based on Llama2. BiomedBERT is tailored for medical applications and excels in extracting essential information from health records, while the Llama2-based chat model facilitates interactions with physicians by providing contextually relevant and reasoning-based answers to their queries. OncoNLP-Assist not only captures the complexity and diversity of clinical language but also provides a robust mechanism for reasoning and answering questions posed by healthcare professionals. This dual capability makes it an invaluable tool in clinical settings, where understanding and responding to nuanced medical queries is crucial. By integrating these advanced NLP techniques, OncoNLP-Assist enhances the efficiency and accuracy of information extraction and decision support, ultimately contributing to improved clinical outcomes and streamlined workflows in oncology treatment care.

## III. DATA COLLECTION AND METHODS

This section presents the clinical text datasets utilized for training our language models and downstream tasks, outlines the BiomedBERT and Llama2 [14] training procedures, and describes the prompt setup for OncoNLP-Assist.

### A. Data

This study uses two sets of free-text reports: the Medical Information Mart for Intensive Care III (MIMIC-III) dataset [15] and a collection of pathology reports from Moffitt Cancer Center. The MIMIC-III dataset is a comprehensive repository of de-identified health data. It includes records from over forty thousand patients admitted to critical care units at Beth Israel Deaconess Medical Center between 2001 and 2012. This dataset is instrumental for training language models due to its extensive coverage of clinical scenarios [15]. Next, the pathology Reports dataset comprises 349,544 pathology reports collected from the Moffitt Cancer Center. These reports span a broad range of cancer diagnoses and treatments,

providing rich, domain-specific information crucial for refining language models to understand oncology-specific contexts. For the masked language model training, each dataset is partitioned into training and evaluation subsets, with 80% of the data allocated for training and 20% for evaluation. Initially, we trained BiomedBERT on the MIMIC-III dataset to familiarize the model with general clinical text. After that, we continue to train the model using the pathology report collection to enhance its understanding of oncology-specific terminologies and treatment protocols. This step-wise training approach ensures that the model gains a robust foundation in the general clinical language before specializing in cancer treatment narratives. Next, we use the i2b2 dataset for downstream tasks to train a named entity recognition model that extracts problems, tests, and treatments. Lastly, for evaluation, we use the Oncology dataset from CaBERT [11], which includes site and histology labels across 2050 patient reports, and compare our results with other available methods.

### B. Bert Training

First, we train a tokenizer from scratch using the WordPiece [16] method, specifically tailored to our pathology report data, resulting in a vocabulary of 32,000 tokens. Next, we start training BiomedBERT from scratch with random initialization and 3 epochs on the MIMIC-III dataset, followed by 3 epochs on the pathology report dataset. The training is distributed across two 40GB Tesla A100 GPUs, with a batch size of 32, a maximum sequence length of 512, and a learning rate set to 5e-05. The entire pre-training process took approximately one day.

After pre-training, the BiomedBERT model is fine-tuned on annotated data to address specific downstream tasks. Our training utilizes the i2b2 dataset for clinical concept extraction, framed as a Named Entity Recognition (NER) task. NER is crucial for information extraction, as it classifies each token in the text into predefined entity classes. This model empowers clinical information retrieval and decision support by extracting and classifying clinical concepts like problems, tests, and treatments.

### C. Chatbot training

**Report Prompt Context**

CaBert extraction:
Primary Site: left breast
Primary Histology: Invasive Ductal Carcinoma

Report:
History of Present Illness: Patient is a 51 year old peri-menopausal female who has no previous complaints or symptoms, presents with a new hard palpable lump in the left breast. {'NEG': ['no'], 'problem': ['previous complaints', 'symptoms', 'a new hard palpable lump in the left breast'], 'body location': ['breast']}
...

Fig. 2. OncoNLP-Assist report prompt example

| Method | F1 score | dataset |
|---|---|---|
| BiomedBERT | 88.5 | 2010 i2b2 |
| Clamp | 88.1 | 2010 i2b2 |
| Amazon Comprehend Medical | 85.5 | 2010 i2b2 |
| CaBERT | 73.2 | Moffitt site |
| DeepPhe | 28.1 | Moffitt site |
| CaBERT | 85.3 | Moffitt hist |
| DeepPhe | 22.4 | Moffitt hist |

TABLE I

EXACT MATCH F1 SCORE OF BIOMEDBERT, CLAMP, AND AMAZON COMPREHEND MEDICAL ON THE NER TASK ACROSS I2B2 2010 CORPORA; CABERT AND DEEPPHE ON THE PREDICTED TUMOR SITE AND HISTOLOGY ACROSS MOFFITT DATASET.

In this study, we fine-tune the Llama2-7B [14] model using low-rank adaptation (LoRA) [17] to enhance its performance as a chatbot. Starting with the Llama2-7B chat model, we leverage its extensive understanding of natural language to provide a robust foundation. The LoRA technique enables us to fine-tune the model efficiently by introducing low-rank updates to the model's weights, significantly reducing the number of trainable parameters and computational resources required. We utilize a pathology report dataset to enhance the model's knowledge in the cancer treatment domain. The training is distributed across two 40GB Tesla A100 GPUs, with 5 epochs, a batch size of 32, an AdamW optimizer [18], and a 5e-05 learning rate.

During inference, we incorporate prompt engineering to provide sufficient context for the model to answer physicians' questions accurately. Figure 2 shows an example report prompt.

## IV. EVALUATION

In this section, we present a comprehensive evaluation of our models for general clinical information extraction using the 2010 i2b2 dataset. We then assess their performance on oncology-related tasks utilizing the Moffitt site and Moffitt hist datasets. Table I compares the performance of our models against notable benchmarks such as Clamp, Amazon Comprehend Medical, and DeepPhe. The evaluation results underscore the efficacy of our BiomedBERT and CaBERT models in extracting and processing clinical information.

For general Clinical Information Extraction, our Biomed-BERT model achieved an F1 score of 88.5 on the 2010 i2b2 dataset, outperforming other models and demonstrating its superior ability to handle clinical information extraction tasks. The Clamp model followed closely with an F1 score of 88.1, and Amazon Comprehend Medical attained an F1 score of 85.5. These results affirm the high effectiveness of our BiomedBERT model in extracting general clinical information. For oncology-related tasks, our models were tested on the Moffitt site and Moffitt hist datasets. The CaBERT model showcased robust performance, achieving an F1 score of 73.2 on the Moffitt site dataset. This indicates its strong capability to extract relevant clinical information from oncology-related data sources. Additionally, CaBERT achieved an F1 score of 85.3 on the Moffitt hist dataset, further validating its

adaptability and precision in handling histopathology-related information. In contrast, the DeepPhe model's performance was notably lower, with F1 scores of 28.1 on the Moffitt site dataset and 22.4 on the Moffitt hist dataset. These results highlight the substantial improvements our models bring to clinical information extraction tasks, particularly in oncology contexts.

Next, we evaluate OncoNLP-Assist using the Moffitt site and hist datasets, focusing on oncology-related questions derived from the provided knowledge base. Given the constraints associated with protected datasets, we are unable to benchmark our chatbot against other advanced models, such as GPT-4 [19] or Gemini [20]. Instead, we compare the performance enhancement imparted by OncoNLP's specialized knowledge to OncoNLP-Assist over the baseline Llama2 chat model. For this evaluation, we employ BERTScore [21], a metric known for its effectiveness in evaluating semantic similarity by leveraging pre-trained BERT models to match words in candidate and reference sentences through cosine similarity. We selected BERTScore due to its robust ability to measure semantic similarity, which is crucial in the medical domain, where information accuracy is paramount.

The results in Table II demonstrate that OncoNLP-Assist consistently outperforms the Llama2 chat model across all evaluated metrics, achieving higher scores in Precision (0.90 ±0.0178 vs. 0.86 ±0.0167), Recall (0.90 ±0.0221 vs. 0.85 ±0.0181), and F1 Score (0.90 ±0.0190 vs. 0.86 ±0.0161). OncoNLP-Assist's higher Precision indicates it retrieves more relevant oncology-related information, reducing the risk of misinformation. Its improved Recall shows it captures a comprehensive set of relevant information crucial for clinical decision-making. The high F1 Score reflects its balanced performance, maintaining both high quantity and quality of relevant clinical data. The consistent improvement across all metrics indicates that OncoNLP-Assist not only captures a wide range of relevant clinical data but also maintains high accuracy in its responses. This enhanced performance is particularly important in medical settings where precision and comprehensiveness are critical for effective decision-making. By outperforming the Llama2 chat model, OncoNLP-Assist proves to be a more reliable tool for oncology-related information retrieval.

Lastly, we evaluate our chatbot's performance against GPT-4 [19], GPT-3.5-turbo [22], and FLAN-UL2 [23], using BLEU [24], ROUGE [25], and Exact Match F1 scores on CORAL dataset [26]. CORAL dataset is a newly developed, fine-grained, expert-labeled dataset of 40 de-identified breast and pancreatic cancer progress notes at the University of California, San Francisco. It includes 20 breast cancer and 20 pancreatic cancer progress notes, featuring 9028 entities, 9986 modifiers, and 5312 relationships. Table III presents the results of zero-shot extraction of detailed oncological information with BLEU-4, ROUGE-1, and exact match (EM) F1-score metrics. Our chatbot outperforms both GPT-3.5-turbo and FLAN-UL2 across all three metrics and demonstrates comparable performance to GPT-4. While GPT-4 slightly

|  | OncoNLP-Assist | Llama2 chat |
|---|---|---|
| Precision | 0.90±0.0178 | 0.86±0.0167 |
| Recall | 0.90±0.0221 | 0.85±0.0181 |
| F1 Score | 0.90±0.0190 | 0.86±0.0161 |

TABLE II

QUANTITATIVE COMPARISON WITH BERTSCORE BETWEEN ONCONLP-ASSIST AND LLAMA2 CHAT.

|  | GPT-4 | GPT-3.5-turbo | FLAN-UL2 | OncoNLP |
|---|---|---|---|---|
| avg BLEU | 0.73 | 0.61 | 0.53 | 0.69 |
| avg ROUGE | 0.72 | 0.58 | 0.27 | 0.59 |
| avg EM-F1 | 0.51 | 0.29 | 0.06 | 0.54 |

TABLE III

COMPREHENSIVE EVALUATION OF ONCONLP-ASSIST, GPT-4, GPT-3.5-TURBO, AND FLAN-UL2 ON THE CORAL DATASET: A COMPARATIVE ANALYSIS OF AVERAGE BLEU, ROUGE, AND EXACT-MATCH F1 SCORES.

outperforms OncoNLP-Assist in BLEU (0.73 vs. 0.69) and ROUGE (0.72 vs. 0.59), OncoNLP-Assist excels in exact match, achieving an EM F1 score of 0.54 compared to GPT-4's 0.51. This highlights OncoNLP-Assist's superior accuracy in precisely matching oncological information, reinforcing its effectiveness as a domain-specific model when compared to general-purpose systems.

## V. DISCUSSION AND FUTURE WORK

In this study, OncoNLP and OncoNLP-Assist have demonstrated significant promise as tools for processing and interacting with oncology data. By fine-tuning these models on specialized oncology datasets and real-world patient pathology reports, we have enhanced their ability to interpret and respond to complex medical information accurately. This approach broadens their application from preliminary patient assessments to automated case adjudication and proactive healthcare measurement, mitigating the risks associated with potential inaccuracies and hallucinations. By employing the OncoNLP pipeline, we aim to enhance OncoNLP-Assist's reliability as a trusted PA, providing it with a well-rounded understanding of patient health conditions and reducing errors and hallucinations.

The modular design of OncoNLP enables upgrades and the integration of new features, ensuring the system remains adaptable and keeps up with the latest medical knowledge. This adaptability is crucial in a rapidly evolving field like oncology, where new research findings and clinical practices continuously emerge. The flexibility of OncoNLP allows for seamless incorporation of these updates, maintaining the tool's relevance and accuracy.

Our future work will focus on expanding the dataset to include more detailed oncology information, further improving the models' performance and utility. By incorporating comprehensive oncology data, including laterality, grade, stage, and lymph node metastasis, we aim to provide a more detailed and nuanced understanding of cancer-related information. This expansion will enable OncoNLP-Assist to deliver more precise and comprehensive insights, supporting a wide range of clinical decision-making processes. Moreover, we plan to enhance OncoNLP-Assist's interaction capabilities, making it more adept at understanding and responding to complex queries from healthcare professionals. By leveraging advanced natural language processing techniques, we aim to create a chatbot that can engage in meaningful, contextually relevant conversations, providing valuable support to clinicians.

## VI. CONCLUSION

In conclusion, the development of OncoNLP and OncoNLP-Assist represents significant advancements in the application of large language models in the medical field, particularly in oncology. This innovative pipeline has demonstrated its potential to assist in various medical tasks, from patient assessment to case adjudication, by leveraging extensive training on real-world medical data. These models' flexibility and accuracy enable them to handle diverse clinical texts and provide contextually relevant responses, enhancing the decision-making process for healthcare professionals. Moreover, OncoNLP's modular design ensures that the system can continuously improve and update with new medical knowledge, keeping it relevant and effective in an ever-evolving field. This adaptability is crucial for maintaining high standards of care and integrating the latest medical advancements. The incorporation of advanced NLP techniques allows OncoNLP-Assist to not only extract essential information from health records but also engage in meaningful interactions with physicians, addressing their queries with precision and depth. Ultimately, OncoNLP and OncoNLP-Assist have the potential to become invaluable tools in healthcare, aiding medical professionals in delivering more accurate diagnoses and treatments. These tools improve the efficiency and accuracy of clinical information extraction and decision support, contributing to better patient outcomes and streamlined healthcare workflows. As the technology continues to evolve, its impact on oncology and other medical specialties is expected to grow, setting the stage for more intelligent and responsive healthcare solutions.

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
