# OpenReview forum: "OncoNLP: Cancer Comprehend Annotation – a pipeline for cancer phenotype and clinical extraction"
_IEEE.org/EMBS/BHI/2024/Conference — IEEE BHI'24_

### Official Review · Reviewer_UJad · 2024-07-24
**Good paper but needs to explain some more details.**

**Overall Rating:** 7
**Confidence:** 4

**Other Quality Metrics:**

(a) Clarity of writing: great
(b) Clinical Significance: good
(c) Methodological Novelty: fair
(d) Experiments and Results: fair

**Questions For The Authors:**

1. The paper mentions using the MIMIC-III and Moffitt Cancer Center datasets for training and evaluation. However, more details are needed on the specific data preprocessing steps taken, such as handling missing values, inconsistencies, data normalization, or biases in the data. Please elaborate on the preprocessing techniques and how they might have impacted the model's performance.
2. The evaluation section primarily focuses on F1 scores and BERTScore for assessing model performance. While these metrics provide insights into accuracy and semantic similarity, they may need to fully capture the clinical relevance and interpretability of the extracted information. How do you plan to address the assessment of clinical validity and utility of the extracted information in future work?
3. The paper introduces OncoNLP-Assist as a chatbot system for interacting with physicians. However, it needs to discuss potential challenges or limitations in real-world clinical settings, such as integrating the system into existing electronic health record (EHR) workflows or ensuring user acceptance and trust. Could you elaborate on strategies to address these implementation challenges and ensure the system's practical usability in clinical practice? Does it need any sort of automation? Will there be any concerning privacy issues?
4. The future work section mentions expanding the dataset to include more detailed oncology information. However, it needs to be clarified how this expansion will be carried out and what specific data sources or annotation strategies will be employed. Could you provide more details on the planned dataset expansion and how it will improve the model's performance and clinical utility? Moreover, it would help if the authors planned to upgrade Llama2 to Llama3.
5. The paper highlights OncoNLP's modular design, which allows for upgrades and the integration of new features. However, it does not discuss the potential for bias or fairness issues that might arise with incorporating new data or models. How do you plan to address and mitigate potential biases in the system to ensure equitable and fair outcomes for all patient populations?

**Strengths:**

The main strength of this paper lies in its innovative approach to addressing the limitations of existing NLP systems in oncology. The development of OncoNLP, a toolkit comprising deep neural network models like BiomedBERT and CaBERT, demonstrates a significant advancement in extracting cancer phenotypes and related biomedical information from pathology reports. The authors' focus on fine-tuning these models on specialized oncology datasets and real-world patient data enhances their ability to interpret complex medical information accurately, making them valuable tools for various medical tasks. Additionally, the modular design of OncoNLP allows for continuous improvement and integration of new features, ensuring its adaptability and relevance in the rapidly evolving field of oncology.

**Summary Of The Paper:**

OncoNLP is a natural language processing (NLP) toolkit designed to extract cancer phenotypes and related biomedical information from pathology reports. It addresses the limitations of existing NLP systems like Amazon Comprehend Medical, Clamp, and DeepPhe, which are proprietary, suboptimal, or need more flexibility in handling diverse clinical narratives. OncoNLP comprises deep neural network BERT models, including BiomedBERT for general medical information and CaBERT for identifying primary tumor sites and histology. It outperforms other methods in extracting clinical information and significantly improves oncology-related tasks. Additionally, OncoNLP-Assist, a chatbot system powered by OncoNLP and Llama2, can extract information from electronic health records and interact with physicians, enhancing the efficiency and accuracy of clinical information extraction and decision support in oncology.

**Weaknesses:**

The evaluation of OncoNLP-Assist, the chatbot system, is a weak point in this paper. The authors mention that they could not benchmark their chatbot against other most advanced models like GPT-4 and Gemini due to constraints associated with protected datasets. Instead, they compared OncoNLP-Assist's performance to the baseline Llama2 chat model. However, the most advanced model, which is far better than Llama2, is currently available - Llama3. While the results showed improvement, the lack of comparison to state-of-the-art models leaves a gap in understanding the capabilities and limitations of OncoNLP-Assist compared to other cutting-edge technologies in the field.

---

### Official Review · Reviewer_mxJK · 2024-08-09
**A whole new toolkit is proposed contributing to the community**

**Overall Rating:** 7
**Confidence:** 4

**Other Quality Metrics:**

a. Excellent

b. Good

c. Fair

d. Good

**Questions For The Authors:**

Why Llama2 was selected instead of another one?

**Strengths:**

S1. A whole new toolkit is proposed contributing to the community.

S2. BiomedBert seems to be better than its competitors.

S3. The authors even present a chatbot system powered by LLMs to assist physicians.

**Summary Of The Paper:**

This paper introduces OncoNLP, an NLP toolkit for extracting cancer phenotypes and related biomedical information. The authors
evaluate the performance of the BiomedBert included in the tool on the i2b2 dataset against Amazon Comprehend Medical and Clamp and show the advantages of the proposed solution. Then they present OncoNLP-Assist, a chatbot system powered by OncoNLP and Llama2 that could extract information from electrical health records and interact with physicians.

**Weaknesses:**

It is not clear why the Llama2 was selected instead of another one.

A discussion is missing on whether such tools would be actually used by clinicians, and maybe a clinician-based evaluation.

Further, given the new AI Act of Europe and similar legislations from the US, a further discussion is missing in this direction.

---

### Official Review · Reviewer_gFtJ · 2024-08-11
**OncoNLP: Cancer Comprehend Annotation – a pipeline for cancer phenotype and clinical extraction**

**Overall Rating:** 7
**Confidence:** 3

**Other Quality Metrics:**

writing: good
significance: great
novelty: good
results: fair

**Questions For The Authors:**

- The datasets used for training and evaluation (MIMIC-III and Moffitt pathology reports) are described, but details on how the data was split and any preprocessing steps are limited. Transparency in these areas is crucial for assessing the robustness and reliability of the results. I would suggest authors elaborate a bit more detail on this.

- I would suggest to re-locate Fig.2. It looks like it's in appendix.

- The future work section discusses expanding the dataset and improving interaction capabilities but lacks specific details on how these improvements will be implemented and evaluated. Please clarify plans for how future datasets will be integrated, how expanded capabilities will be developed, and how these changes will be validated. This would provide a more complete picture of the project's roadmap.

**Strengths:**

- The modular design of OncoNLP allows for easy upgrades and integration of new models, making it scalable to evolving needs and improvements in NLP technologies.

- Use of Low-Rank Adaptation (LoRA) for fine-tuning the Llama2 model efficiently, which enhances performance while reducing computational resources and training time. The approach allows the Llama2 model to be fine-tuned on domain-specific data (cancer reports), improving its ability to assist with cancer-related queries.

- For the chatbot component, prompt engineering is utilized to ensure the model provides contextually relevant and accurate answers to physicians' queries, enhancing the usability and effectiveness of the system in real-world scenarios.

**Summary Of The Paper:**

The submitted work introduces OncoNLP, a new natural language processing (NLP) toolkit designed for extracting cancer-related information from clinical texts. The toolkit includes two primary components. BiomedBERT: A model for extracting general medical information. CaBERT: A model specifically for identifying the primary tumor site and histology. The performance of BiomedBERT was evaluated using the i2b2 dataset and was found to outperform other methods like Amazon Comprehend Medical and Clamp, achieving an F1-score of 88.5%. CaBERT was tested on the Moffitt dataset and showed a 50% improvement over DeepPhe in tasks related to tumor site and histology identification. In addition to OncoNLP, OncoNLP-Assist, an advanced chatbot system that combines OncoNLP with the Llama2 to interact with physicians is also developed, extracting information from electronic health records.

**Weaknesses:**

- The evaluation of OncoNLP-Assist is only compared against the Llama2 chat model, rather than other advanced models like GPT-4 or Gemini. This limits the understanding of how OncoNLP-Assist performs relative to the most recent and sophisticated NLP models.

- While the F1 scores provided for BiomedBERT and CaBERT are impressive, they may not fully capture the model's performance in practical clinical settings. The benchmarks are limited to a few models and do not cover all relevant aspects. Additional metrics such as precision, recall could provide a more comprehensive evaluation of the models' strengths and weaknesses.

- Although the work highlights security issues with online services, it does not provide an in-depth analysis of how OncoNLP and OncoNLP-Assist handle sensitive data securely. In healthcare, ensuring data privacy and security is critical. Providing information on how the system adheres to regulations such as HIPAA, and any specific security measures in place, would strengthen the work’s credibility and applicability in real-world scenarios, I believe.

---

### Decision · Program_Chairs · 2024-09-23

Accept